# Tobramycin Promotes Melanogenesis by Upregulating p38 MAPK Protein Phosphorylation in B16F10 Melanoma Cells

**DOI:** 10.3390/antibiotics8030140

**Published:** 2019-09-05

**Authors:** Seung-Hyun Moon, You Chul Chung, Chang-Gu Hyun

**Affiliations:** Department of Chemistry and Cosmetics, Jeju National University, Jeju 63243, Korea (S.-H.M.) (Y.C.C.)

**Keywords:** melanogenesis, B16F10 melanoma cell, aminoglycoside, tobramycin, melanin, p-p38, kanamycin, 2-DOS

## Abstract

Tobramycin is an aminoglycoside-based natural antibiotic derived from *Streptomyces tenebrarius*, which is primarily used for Gram-negative bacterial infection treatment. Although tobramycin has been utilized in clinical practice for a long time, it has exhibited several side effects, leading to the introduction of more effective antibiotics. Therefore, we conducted our experiments focusing on new possibilities for the clinical use of tobramycin. How tobramycin affects skin melanin formation is unknown. This study used B16F10 melanoma cells to assess the effect of tobramycin on melanin production. After cytotoxicity was assessed by MTT assay, melanin content and tyrosinase activity analyses revealed that tobramycin induces melanin synthesis in B16F10 cells. Next, Western blot analyses were performed to elucidate the mechanism by which tobramycin increases melanin production; phosphorylated p38 protein expression was upregulated. Protein inhibitors have been used to elucidate the mechanism of tobramycin. Kanamycin A and B are structurally similar to tobramycin, and 2-DOS represents the central structure of these antibiotics. The effects of these substances on melanogenesis were evaluated. Kanamycin A reduced melanin production, whereas kanamycin B and 2-DOS had no effect. Overall, our data indicated that tobramycin increases melanin production by promoting p38 protein phosphorylation in B16F10 melanoma cells.

## 1. Introduction

Ultraviolet radiation (UVR) is a mutagen and a nonspecific damaging agent, classified as a complete carcinogen, since it exhibits both tumor initiator and tumor promoter properties. UVR also causes various acute and chronic skin reactions. Acute human skin reactions to UVR include photodamage, erythema, mutation, and immunosuppression [1,2].

Melanin is produced by the melanocytes present in the skin to prevent UV damage. First, UV-mediated DNA damage in keratinocytes activates p53 and binds to and activates the pro-opiomelanocortin (POMC) gene. POMC polypeptides are post-translationally cleaved to produce adrenocortical stimulating hormones, α-melanocyte-stimulating hormones (α-MSH), and β-endorphins. The secreted keratinocyte-derived α-MSH signals to melanocytes through the G protein-coupled receptor melanocortin 1 receptor (MC1R) [3]. α-MSH binds to the melanocyte MC1R and stimulates the adenylyl cyclase (AC) to increase the intracellular concentration of the secondary messenger cyclic adenosine monophosphate (cAMP). cAMP activates protein kinase A (PKA) and phosphorylates the cAMP response element (CREB) protein. In turn, CREB acts as a transcription factor for several genes, including the microphthalmia-associated transcription factor (MITF), a major regulator of melanogenesis. MITF binds to the M-box and regulates the expression of the melanin-producing enzymes tyrosinase (TYR), tyrosinase related protein 1 (TRP-1), and tyrosinase related protein 2 (TRP-2).

Recent studies have revealed signaling pathways affecting the formation of various melanins, such as the phosphoinositide 3-kinase (PI3K)/protein kinase B (AKT), Wnt/β-catenin, and mitogen-activated protein kinases (MAPK) signaling pathways; the MAPK pathway includes the extracellular signal-regulated kinase (ERK), c-Jun N-terminal kinase (JNK), and p38. It is also known that diverse factors, such as SCF/c-kit, nitric oxide (NO), and cytokines, inhibit or promote melanogenesis [3,4,5].

TYR and TRP-1,2 proteins migrate to the organelles called melanosomes, which produce two types of melanin (eumelanin and pheomelanin) [6]. Thus, melanin is formed in the skin, hair, and eyes; it not only protects the skin from UV, but also determines skin color [7,8]. Melanin, especially eumelanin, exerts a shielding effect and acts as a physical barrier scattering UVR and as an absorbent filter reducing UV penetration into the epidermis. Melanin efficacy as a sunscreen is assumed to be about 1.5–2.0 sun protection factor (SPF), and can reach up to 4 SPF, meaning that melanin absorbs 50% to 75% of the UVR [9].

Hypopigmentation in the skin is the result of melanin production reduction. This is due to a decrease in the amino acid tyrosine caused by melanin cell number reduction, melanin deficiency, or defects in melanin use by melanocytes. Common genetic causes of hypopigmentation include mutations in the tyrosinase gene. Symptoms encompass vitiligo, piebaldism, and albinism; various studies are under way to identify improved treatment options [10,11,12]. 

Tobramycin (Figure 1) is an aminoglycoside antibiotic with broad antibacterial spectrum in vitro, which is primarily used for treating serious Gram-negative bacterial infections [13]. Tobramycin is a natural compound, produced in culture broth of *Streptomyces tenebrarius*, and it is composed of a 2,6-diamino-2,6-dideoxy sugar and a 3-amino-3-deoxyglucose linked to the 4- and 6-hydroxyl groups of 2-deoxystreptamine [14,15]. It was discovered in 1967 and is currently still in clinical use [16,17,18,19]. However, tobramycin exhibits side effects such as kidney damage, hearing damage, and serious allergic reactions [15,20,21,22]. Over time, its use has gradually decreased due to the emergence of antibiotic-resistant bacteria and the development of more effective antibiotics [23,24,25]. Therefore, we initiated the current study to identify other possible clinical uses of tobramycin.

## 2. Results

### 2.1. Cell Viability of B16F10 Melanoma Cells

The 3-(4,5-dimethylthiazol-2-yl)-2,5-diphenyltetrazolium bromide (MTT) assay was conducted to determine tobramycin cytotoxicity on B16F10 cells. The cells were cultured for 24 h, treated with various concentrations of tobramycin, cultured for further 72 h, and supplemented with the MTT reagent for cell viability measurement. Cytotoxicity was observed at 8 mg/mL concentration of tobramycin, and further experiments proceeded at a concentration of 4 mg/mL of tobramycin. 

To evaluate if the selected concentrations of tobramycin could cause cellular damage and cytolysis, the lactate dehydrogenase (LDH) released to the extracellular medium was spectrophotometrically quantified. It was observed that tobramycin 8 and 16 mg/mL induced LDH release (60% and 400% variation, respectively, to untreated cell), while other concentrations of tobramycin did not alter LDH release relative to the untreated cell. Due to cellular toxicity induced by the high concentrations of tobramycin (8 and 16 mg/mL), these concentrations were not considered for the present study (Figure 2).

### 2.2. Effect of Tobramycin on Melanin Synthesis

Melanin protein levels, produced in B16F10 cells after treatment with tobramycin at various concentrations (0.5, 1, 2, and 4 mg/mL), were measured spectrophotometrically. In the cells treated with tobramycin for 72 h, melanin production was increased in a concentration-dependent manner (Figure 3a). Figure 3b shows melanin accumulation in the cells; as the concentration of tobramycin increases, the melanin synthesis becomes macroscopically detectable.

### 2.3. Effect of Tobramycin on Tyrosinase Activity

Melanin synthesis occurs in the melanosomes, and it is initiated with tyrosine oxidation by tyrosinase. Therefore, we measured the activity of tyrosinase, an enzyme important for melanin synthesis, in tobramycin-treated B16F10 cells. Tyrosinase activity was increased in tobramycin-treated cells compared to untreated cells (Figure 4).

### 2.4. Expression of Proteins Related to Melanin Synthesis

#### 2.4.1. Tyrosinase, TRP-1, TRP-2

Tyrosinase, TRP-1, and TRP-2 are important enzymes in melanin synthesis. Tobramycin stimulated melanin synthesis, and the effect of tobramycin on the intracellular expression of these three proteins was confirmed by Western blotting (Figure 5). Tyrosinase, TRP-1, and TRP-2 levels increased with higher tobramycin concentrations.

#### 2.4.2. MITF

MITF is a protein regulating tyrosinase, TRP-1, and TRP-2 expression. Therefore, the effect of tobramycin on MITF levels was evaluated. MITF expression increased in B16F10 cells treated with tobramycin for 20 h in a dose-dependent manner (Figure 6). These data imply that tobramycin increases MITF expression, which in turn upregulates tyrosinase, TRP-1, and TRP-2 levels.

#### 2.4.3. MAPKs and AKT

The MAPK signaling pathway and the PI3K/AKT signaling pathway have been implicated in regulating MITF expression. MAPK family proteins, including ERK1/2, JNK, and p38, play an important role in melanin synthesis. Recent studies have shown that ERK phosphorylation degrades MITF by ubiquitination and inhibits TYR and TRP-1,2 expression, whereas JNK and p38 phosphorylation increases MITF expression, thereby inducing TYR and TRP-1,2 upregulation [26,27,28]. In addition, phosphorylated AKT has been implicated in melanin synthesis inhibition by phosphorylating MITF [29]. p38 phosphorylation levels tended to increase with higher concentrations of tobramycin in Western blot experiments. However, ERK, JNK, and AKT expression did not change in a dose-dependent manner (Figure 7 and Figure 8).

### 2.5. Tyrosinase Activity Assay with Protein Inhibitors

Tyrosinase activity increased when measured with an ERK inhibitor and a JNK inhibitor, but it decreased when a p38 inhibitor was used. These data confirm the notion that tobramycin phosphorylates p38, thereby increasing melanin formation (Figure 9). In addition, H89, a PKA inhibitor, was used to determine the effect of tobramycin on the cAMP/PKA signaling pathway (Figure 10). PKA activation increased MITF expression. Should tobramycin affect the cAMP/PKA signaling pathway, tyrosinase activity would be reduced using an PKA inhibitor; however, tyrosinase activity did not decrease with PKA inhibitor treatment. Tobramycin showed an increased cAMP/PKA signal. These data confirm that PKA is not involved in the effect of tobramycin on melanin synthesis.

### 2.6. Effects of Tobramycin-Related Structures on Melanogenesis

The cytotoxic effect of kanamycin A, kanamycin B, and 2-DOS, which share an antibiotic skeleton and similar structure with tobramycin, was measured (Figure 11). Subsequently, melanin content experiments were performed at concentration range showing no cytotoxicity. Kanamycin A demonstrated a tendency to decrease melanin content, whereas kanamycin B and 2-DOS did not change melanin content (Figure 12 and Figure 13).

### 2.7. Cell Viability of HaCaT Keratinocyte Cells

Both melanocytes and keratinocytes are present in the basal layer of the skin; furthermore, protrusions, extending from each melanocyte, are in contact with 30 to 40 peripheral keratinocytes. To confirm the effect of tobramycin on the keratinocytes surrounding melanocytes, an MTT assay was performed using a human cell line (HaCaT). Keratinocytes showed cell viability of 83% at 4 mg/mL tobramycin, in contrast to melanocytes, in which cell viability was significantly reduced at 8 mg/mL tobramycin (Figure 14).

## 3. Discussion

Melanin, which protects the skin against UVR, is produced in melanosomes and is present in the human skin, hair, eyes, ears, and even brain [30,31]. Excessive pigmentation causes hyperpigmentation, whereas insufficient pigmentation is associated with the degradation of pigmentation [6]. Here, we studied the effect of tobramycin, one of the aminoglycoside antibiotics, on melanogenesis. Tobramycin has been in clinical use for a long time; however, it has demonstrated several side effects and has increasingly been replaced by improved antibiotics. Here, we demonstrated the tobramycin effect on melanin formation in B16F10 cells. First, a MTT assay was performed to evaluate tobramycin cytotoxicity. It confirmed that tobramycin treatment increases the activity of tyrosinase, which is the most important enzyme for melanin synthesis. Furthermore, tobramycin upregulated melanin levels in a concentration-dependent manner at a concentration range which did not cause cytotoxicity. We identified the effects of tobramycin on the cAMP/PKA, MAPK, and PI3K/AKT signaling pathways, which are important pathways for melanogenesis, using Western blot experiments and the tyrosinase activity assay with protein inhibitors. First, the expression levels of tyrosinase, TRP-1, and TRP-2, the most relevant enzymes for melanin synthesis, were increased when cells were treated with tobramycin. MITF is a factor known for controlling the levels of these three enzymes; its expression was increased in B16F10 cells treated with tobramycin for 20 h in a dose-dependent manner. The mechanism involved in MITF expression included tobramycin phosphorylation of the p38 protein, one of the MAPK proteins, among various signal transduction pathways. Phosphorylated p38 increased MITF expression, which in turn upregulated tyrosinase, TRP-1, and TRP-2 levels, leading to heightened melanin synthesis.

The structure of the aminoglycoside antibiotics includes a central skeleton, which is an aminocyclitol ring with an amino sugar bound to it. Kanamycin A, kanamycin B, and tobramycin incorporate the central structure presented by 2-DOS. Furthermore, kanamycin A, kanamycin B, and tobramycin contain a structure in which an amino sugar is bonded to carbons 4 and 6. The effect of the basic skeleton structure, 2-DOS, on melanin synthesis (Figure 11 and Figure 12) was measured in a concentration range not causing cytotoxicity (Figure 11). Treatment with 2-DOS did not affect melanogenesis, suggesting that the 2-DOS structure does not influence tobramycin-induced melanogenesis. Tobramycin is a derivative of kanamycin, which has a very similar structure. Kanamycin A, kanamycin B, and tobramycin decreased, did not influence, and improved melanin production, respectively [32]. Structural comparison of the three drugs shows that the structure of the amino sugar bound to the 4th carbon of 2-DOS is slightly different. Kanamycin A binds to 2’-OH, 3’-OH, 4’-OH, 5’-CH2NH2, whereas kanamycin B binds to 2’-NH2, 3’-OH, 4’-OH, 5’-CH2NH2, and tobramycin binds to 2’-NH2, 4’-OH, 5’-CH2NH2 (Figure 15). The data imply that tobramycin effect on melanogenesis is due to the structural difference of the amino sugars bound to the 4-carbon of 2-DOS.

Melanocytes are uniformly distributed throughout the basal layer of the skin epidermis [33]. Melanin, produced from melanocytes, migrates to surrounding keratinocytes. Due to the close relationship between melanocytes and keratinocytes, tobramycin cytotoxicity was evaluated on keratinocytes (Figure 12). The cytotoxicity of tobramycin in HaCaT human keratinocytes was 83% and more than 90% after 4 mg/mL and 2 mg/mL tobramycin treatment, respectively. Even though tobramycin demonstrated slight toxicity at 4 mg/mL in HaCaT human keratinocytes, it could be used at lower concentrations. We have confirmed from the above experiments that Tobramycin has the effect of improving melanin synthesis in melanocytes, apart from the role of antibiotics. Additional experiments will be needed for medical use but we believe this has proven the new efficacy of this old antibiotic, which is no longer in use.

## 4. Materials and Methods 

### 4.1. Chemicals and Reagents 

Tobramycin and kanamycin A were purchased from Tokyo Chemical Industry Co., Ltd. (Chuo-ku, Tokyo, Japan). Dulbecco’s Modified Eagle Medium (DMEM), fetal bovine serum (FBS), penicillin/streptomycin, trypsin-ethylenediaminetetraacetic acid, PD98059, and BCA kit were procured from Thermo Fisher Scientific (Waltham, MA, USA). α-MSH, NaOH, MTT, and kanamycin B were obtained from Sigma-Aldrich (St. Louis, MO, USA). Antibodies against tyrosinase, TRP-1, TRP-2, and MITF were obtained from Santa Cruz Biotechnology (Dallas, TX, USA). Antibodies against p-p38, p38, p-JNK, JNK, p-ERK, ERK, p-AKT, AKT, and β-actin were purchased from Cell Signaling Technology (Danvers, MA, USA). Radioimmunoprecipitation assay (RIPA) buffer, dimethyl sulfoxide (DMSO), enhanced chemiluminescence (ECL) kit, and 2× Laemmli sample buffer were obtained from Biosesang (Sungnam, Gyeonggi-do, Korea). SP600125 and SB203580 were purchased from Cayman Chemical (Ann Arbor, MI, USA) and Calbiochem (San Diego, CA, USA); 2-DOS was obtained from Toronto Research Chemicals Inc. (Toronto, Ontario, Canada). EZ-LDH cell cytotoxicity assay kit was purchased from DoGenBio Co.,Ltd (Guro, Seoul, Korea).

### 4.2. Cell Culture

B16F10 murine melanoma cells and HaCaT keratinocyte cells were cultured in DMEM medium supplemented with 10% heat inactivated FBS and 1% penicillin/streptomycin at 37 °C in a humidified atmosphere containing 5% CO_2_.

### 4.3. Cell Viability Assay

Cell viability was determined by MTT assay. B16F10 melanoma cells (3.0 × 10^4^ cells/well) and HaCaT keratinocyte cells (1 × 10^4^ cells/well) were seeded into 24-well plates and incubated for 24 h in culture medium. The cells were treated with various concentrations of tobramycin (0.5, 1, 2, 4, 8, and 16 mg/mL), kanamycin A (0.25, 0.5, 1, 2, and 4 mg/mL), kanamycin B (0.125, 0.25, 0.5, 1, and 2 mg/mL), and 2-DOS (0.125, 0.25, 0.5, 1, and 2 mg/mL) for 24 h (HaCaT) or 72 h (B16F10). After incubation, MTT solution was added to a final concentration of 0.2 mg/mL for 2 h. Next, the MTT solution was removed, the produced formazan crystals were dissolved by DMSO, and absorbance was measured at 540 nm on a spectrophotometric microplate reader (Tecan, Mannedorf, Switzerland).

### 4.4. Lactate Dehydrogenase (LDH) Release Assay

B16F10 cells were seeded in a 24-well plate and cultured 24 h and incubated with various concentrations of tobramycin (8 and 16 mg/mL) for 72 h. After incubation, 10 μL of the culture supernatant was transferred into a new 96-well plate. LDH released to the extracellular medium was quantified by the EZ-LDH cell cytotoxicity assay kit according to the manufacturer’s instructions. Fresh culture medium was used as blank. The absorbance was measured at 450 nm using an ELISA reader.

### 4.5. Melanin Content Measurement

B16F10 cells (1.0 × 10^5^ cells/well) were seeded into 60π dish and incubated for 24 h in culture medium. Next, the medium was removed, and the cells were treated with various tobramycin (0.5, 1, 2, and 4 mg/mL), kanamycin A (0.5, 1, and 2 mg/mL), kanamycin B (0.0625, 0.125, and 0.25 mg/mL), and 2-DOS (0.0625, 0.125, 0.25 mg/mL) concentrations, 100 nM or 200 nM α-MSH, and 200 nM arbutin for 72 h. After incubation, the cells were washed with PBS once, and were dissolved in 2 mL 1 N NaOH containing 10% DMSO at 70 °C for 1 h. Then, 200 μL NaOH solution aliquots were transferred to 96-well plates, and absorbance was measured at 405 nm on a spectrophotometric microplate reader (Tecan, Mannedorf, Switzerland). The NaOH solution protein concentration was quantified by a bicinchoninic acid (BCA) protein assay kit.

### 4.6. Intracellular Tyrosinase Activity

B16F10 cells (1.0 × 10^5^ cells/well) were seeded into a 60π dish and incubated for 24 h in culture medium. Next, the medium was removed, and the cells were treated with various tobramycin (0.5, 1, 2, and 4 mg/mL), kanamycin A (0.5, 1, and 2 mg/mL), kanamycin B (0.0625, 0.125, and 0.25 mg/mL), and 2-DOS (0.0625, 0.125, and 0.25 mg/mL) concentrations, 100 nM or 200 nM α-MSH, 200 nM arbutin, and protein inhibitors for 72 h. After incubation, the cells were washed with PBS once, and were treated with RIPA buffer (supplemented with 1% protease inhibitor cocktail) for 20 min at 4 ℃. Cell lysates were collected with cell scrapers and transferred to 1.5 mL microtubes. Collected lysates were centrifuged at 15,000× *g* for 25 min, and the supernatant was separated. Protein concentration was quantified by BCA protein assay kit. The amount of quantified protein in the samples was equalized using 0.1 M sodium phosphate buffer; the samples were then mixed with 2 mg/mL L-DOPA per well in 96-well plates. After incubation at 37 °C and centrifugation at 300 rpm for 1 h, absorbance was measured at 490 nm on a spectrophotometric microplate reader. 

### 4.7. Western Blot Analysis

B16F10 cells (1.0 × 10^5^ cells/dish) were seeded into a 60π dish, incubated, and then treated with various concentrations of tobramycin (0.5, 1, 2, and 4 mg/mL) for predetermined periods of time to analyze the expression of proteins of interest. After incubation, the cells were collected and lysed in RIPA buffer (supplemented 1% protease inhibitor cocktail). Next, collected lysates were centrifuged at 15,000× *g* for 25 min, and the supernatant was separated. Protein concentration in the supernatant was quantified by BCA protein assay kit. Supernatant and 5ⅹ SDS-PAGE buffer (4:1) were mixed to prepare Western blotting samples (protein concentration: 40 μg). Finally, samples (10 μL) were loaded on sodium dodecyl sulfate-polyacrylamide gels.

### 4.8. Statistical Analysis

The results of the experiments were analyzed using Student’s *t*-test. *p*-values < 0.05 (*) or 0.01 (**, ##) or 0.001 (***, ###) were considered statistically significant and all data are expressed as percentages compared to the respective values of the control or negative control. All data has been expressed as mean ± SD of at least three independent experiments.

## Figures and Tables

**Figure 1 antibiotics-08-00140-f001:**
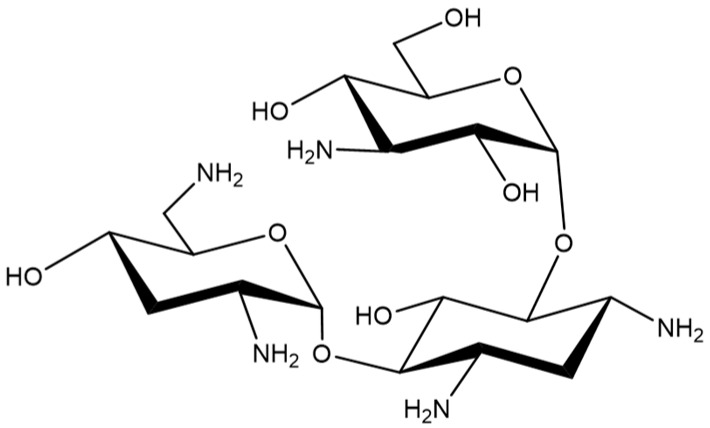
Chemical structure of Tobramycin.

**Figure 2 antibiotics-08-00140-f002:**
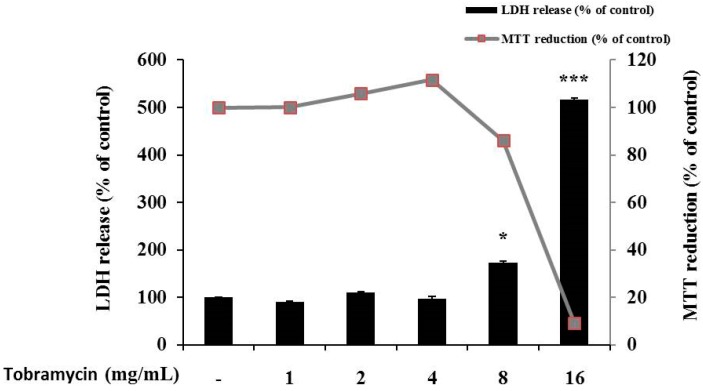
Cell viability of tobramycin-treated B16F10 melanoma cells. The cells were supplemented with various concentrations of tobramycin for 72 h. Data are presented as mean ± standard deviation (SD) of at least four independent experiments (*n* = 4). ** indicates *p* < 0.01, *** *p* < 0.001 vs. untreated cells.

**Figure 3 antibiotics-08-00140-f003:**
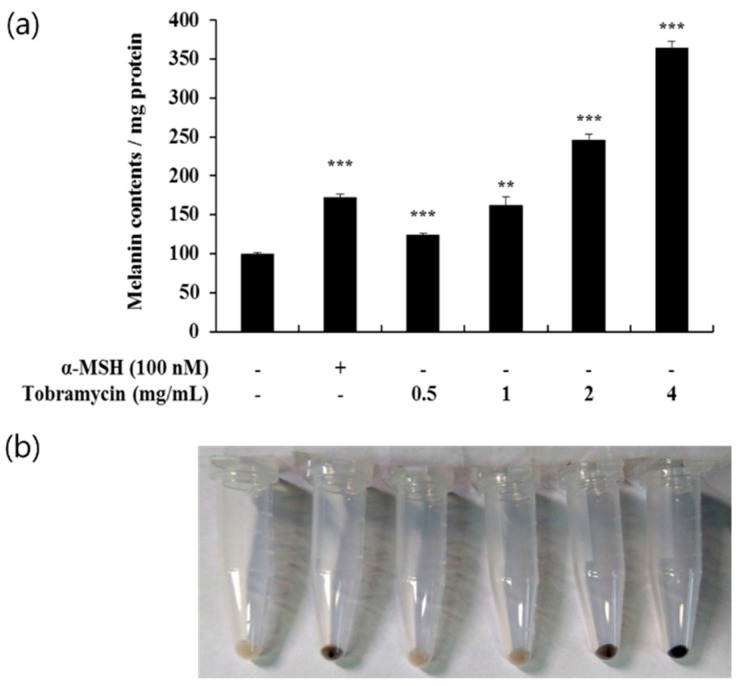
Melanin levels in tobramycin-treated B16F10 melanoma cells. The cells were treated with various concentrations of tobramycin for 72 h, and α-MSH was used as a positive control. (**a**) Melanin concentrations are expressed as percentages compared to the respective values obtained for the control cells. (**b**) Images of corresponding B16F10 cell pellets harvested by centrifugation. Data are presented as mean ± standard deviation (SD) of at least four independent experiments (*n* = 4). *** indicates *p* < 0.001 vs. untreated cells.

**Figure 4 antibiotics-08-00140-f004:**
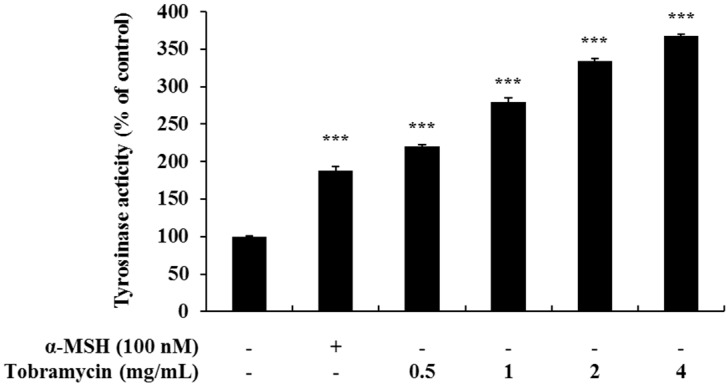
Tyrosinase activity in tobramycin-treated B16F10 melanoma cells. The cells were treated with various concentrations of tobramycin for 72 h, and α-MSH was used as a positive control. Data are presented as mean ± standard deviation (SD) of at least four independent experiments (*n* = 4). *** indicates *p* < 0.001 vs. untreated cells.

**Figure 5 antibiotics-08-00140-f005:**
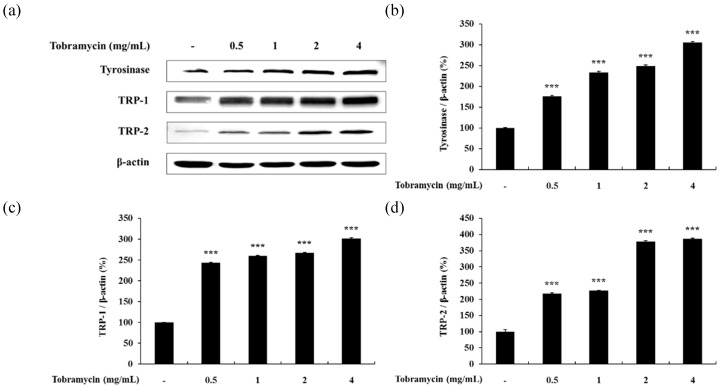
Effect of tobramycin on tyrosinase, TRP-1, and TRP-2 expression in B16F10 cells. Cells were treated with various concentrations of tobramycin for 40 h. Protein levels were examined by Western blotting. (**a**) Representative Western blotting results and quantified (**b**) tyrosinase, (**c**) TRP-1, and (**d**) TRP-2 protein levels. Results are expressed as percentages of the control. Data are presented as mean ± standard deviation (SD) of at least three independent experiments (*n* = 3). *** indicates *p* < 0.001 vs. untreated cells.

**Figure 6 antibiotics-08-00140-f006:**
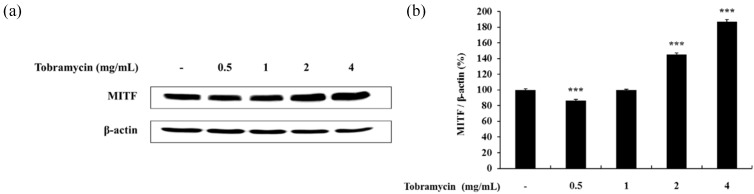
Effect of tobramycin on MITF expression in B16F10 cells. Cells were treated with various concentrations of tobramycin for 20 h, and protein levels were examined by Western blotting. (**a**) Representative Western blotting results, and (**b**) quantified MITF protein levels. Results are expressed as percentages of the control. Data are presented as mean ± standard deviation (SD) of at least three independent experiments (*n* = 3). *** indicates *p* < 0.001 vs. untreated cells.

**Figure 7 antibiotics-08-00140-f007:**
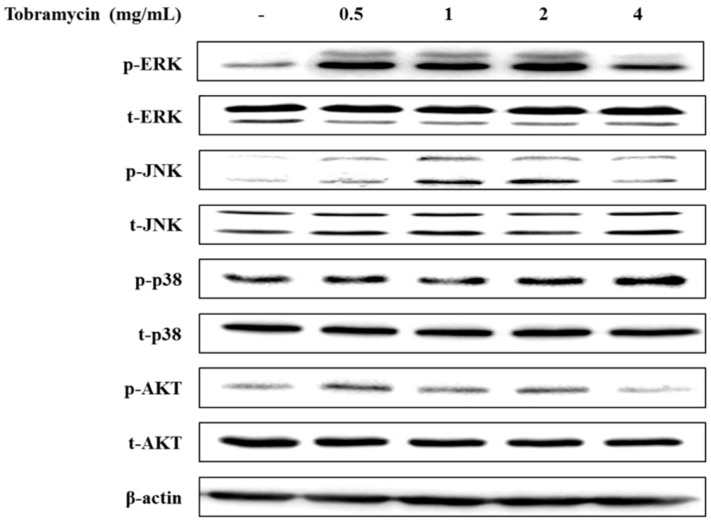
Effect of tobramycin on MAPK expression in B16F10 cells. Cells were treated with various concentrations of tobramycin for 4 h. Protein levels were examined by Western blotting.

**Figure 8 antibiotics-08-00140-f008:**
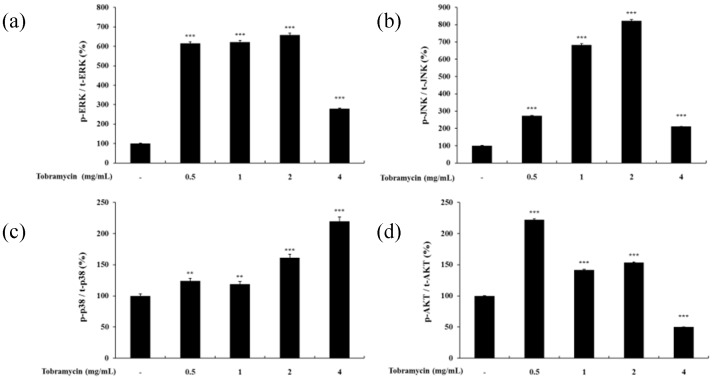
Quantified protein levels of (**a**) p-ERK, (**b**) p-JNK, (**c**) p-p38, and (**d**) p-AKT from Western blot experiments. Results are expressed as percentages of the control. Data are presented as mean ± SD of at least three independent experiments (*n* = 3). ** indicates *p* < 0.01, *** *p* < 0.001 vs. control.

**Figure 9 antibiotics-08-00140-f009:**
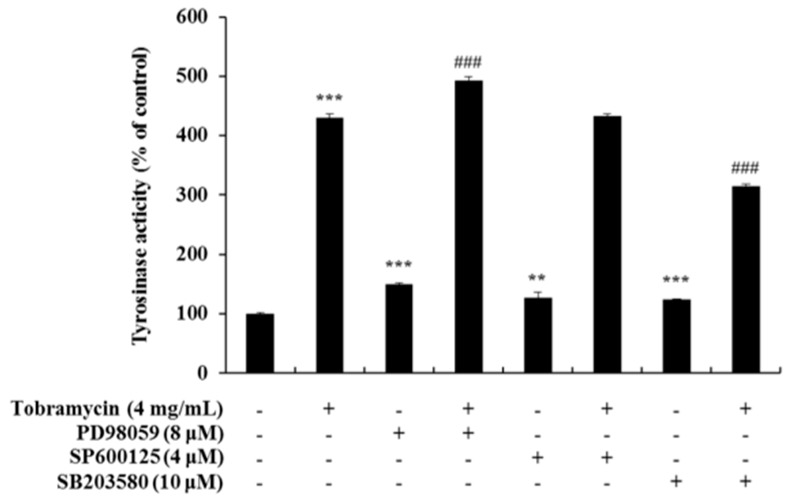
Effect of MAPK inhibitors on tobramycin-induced tyrosinase activity in B16F10 cells. To confirm the mechanism underlying the effect of tobramycin in melanogenesis, cellular tyrosinase activity was measured using the following MAPK inhibitors: PD98059 (ERK inhibitor), SP600125 (JNK inhibitor), and SB203580 (p38 inhibitor). Results are expressed as percentages of the control. Data are presented mean ± SD of four independent experiments (*n* = 4). ** indicates *p* < 0.01 *** indicates *p* < 0.001 vs. untreated cells and ### indicates *p* < 0.001 vs. tobramycin treated cells.

**Figure 10 antibiotics-08-00140-f010:**
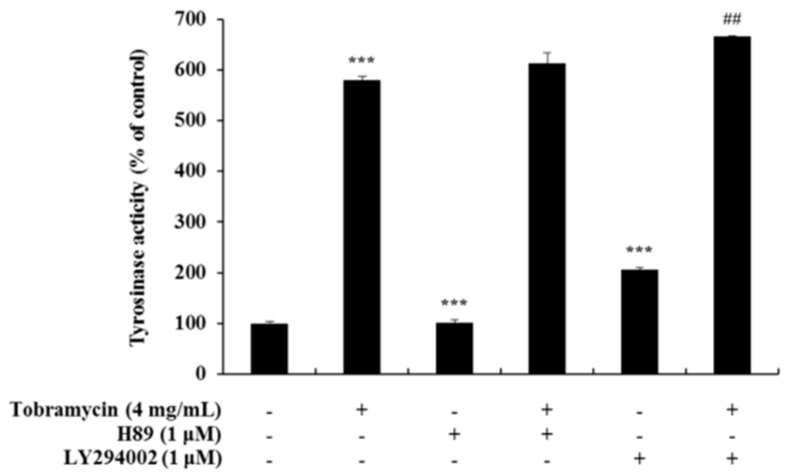
Effect of PKA inhibitors on tobramycin-induced tyrosinase activity in B16F10 cells. To confirm tobramycin mechanism in melanogenesis, cellular tyrosinase activity was measured using H89 (PKA inhibitor) and LY294002 (AKT inhibitor). Results are expressed as percentages of the control. Data are presented as mean ± SD of four independent experiments (*n* = 4). *** indicates *p* < 0.001 vs. untreated cells and ## indicates *p* < 0.01 vs. tobramycin treated cells.

**Figure 11 antibiotics-08-00140-f011:**
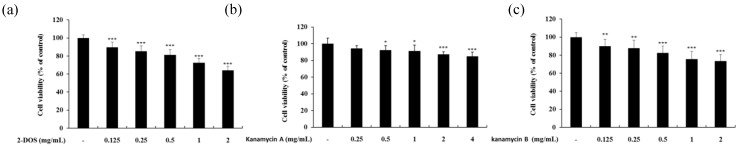
Cell viability of kanamycin A-(**a**), kanamycin B-(**b**), and 2-DOS-(**c**) treated B16F10 melanoma cells. Cells were treated with various concentrations of these drugs for 72 h. Data are presented as mean ± standard deviation (SD) of at least four independent experiments (*n* = 4). * indicates *p* < 0.05, ** *p* < 0.01, *** *p* < 0.001 vs. untreated cells.

**Figure 12 antibiotics-08-00140-f012:**
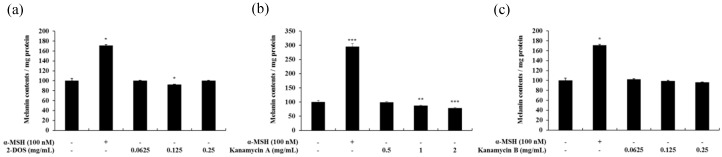
Melanin contents in 2-DOS-(**a**), kanamycin A-(**b**), and kanamycin B-(**c**) treated B16F10 melanoma cells. The cells were treated with various concentrations for 72 h. α-MSH was used as a positive control. Melanin concentrations are expressed as percentages compared to the respective values obtained for the control cells. Data are presented as mean ± standard deviation (SD) of at least four independent experiments (*n* = 4). * indicates *p* < 0.05, ** *p* < 0.01, *** *p* < 0.001 vs. untreated cells.

**Figure 13 antibiotics-08-00140-f013:**
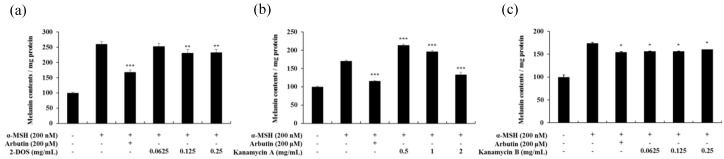
Melanin contents of 2-DOS(**a**), kanamycin A(**b**), and kanamycin B(**c**)-treated B16F10 melanoma cells. The cells were treated with various drug concentrations for 72 h. α-MSH was used as a negative control and arbutin was used as a positive control. Melanin concentrations are expressed as percentages compared to the respective values obtained for the control cells. Data are presented as mean ± standard deviation (SD) of at least four independent experiments (*n* = 4). * indicates *p* < 0.05, ** *p* < 0.01, *** *p* < 0.001 vs. tobramycin treatment.

**Figure 14 antibiotics-08-00140-f014:**
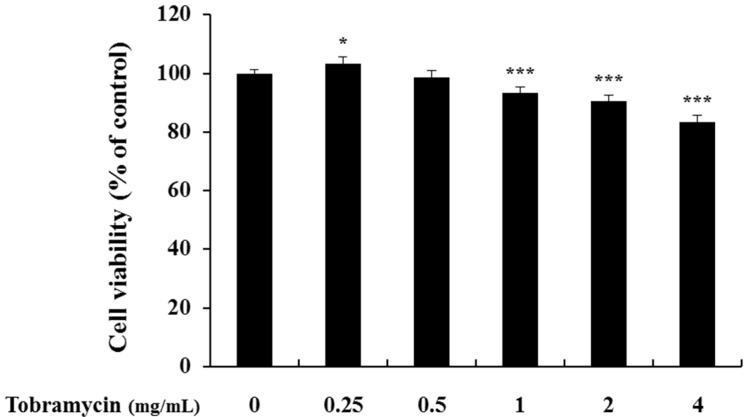
Cell viability of tobramycin-treated HaCaT keratinocyte cells. The cells were treated with various concentrations of tobramycin for 24 h. Data are presented as mean ± standard deviation (SD) of at least four independent experiments (*n* = 4). * indicates *p* < 0.05, *** *p* < 0.001 vs. control.

**Figure 15 antibiotics-08-00140-f015:**
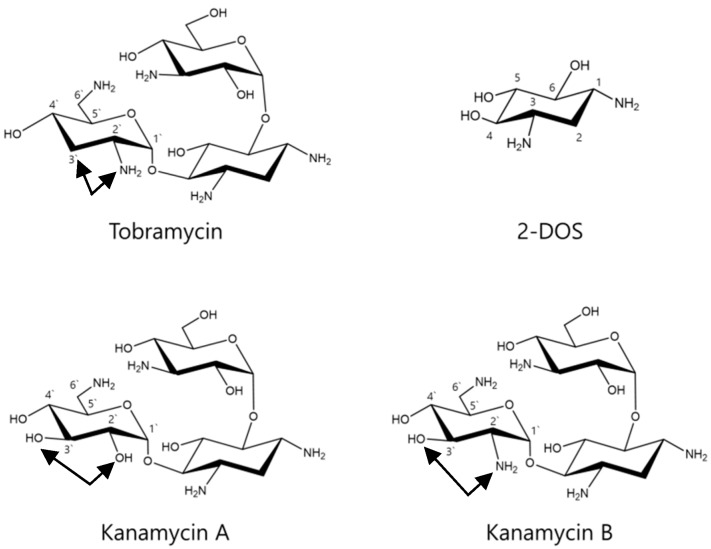
Structures of 2-DOS, tobramycin, kanamycin A, and kanamycin B.

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
