# Peer review of "Tobramycin Promotes Melanogenesis by Upregulating p38 MAPK Protein Phosphorylation in B16F10 Melanoma Cells"

_antibiotics, 2019, doi:10.3390/antibiotics8030140_

Round 1

Reviewer 1 Report

            Looking to re-purpose the venerable antibiotic tobramycin, the authors test it’s activity in a melanoma cell line and find that it increased melanin production. They carefully trace this activity to the so-called melanogenesis master regulator MITF, and further upstream to a MAPK pathway with phosphorylated P38. They show that these transformed melanocytes have good viability with tobramycin, and keratincytes have reasonable viability at an effective dose for inducing melanogenesis. Thus, tobramycin could potentially be re-purposed for therapeutic and cosmetic skin applications. A comparison with structurally related aminoglycoside antibiotics suggests they don’t have a common target (or they affect a common target in different ways) and that their impact on melanogenesis is quite structurally specific (tobramycin and kanamycin B vary by a single hydroxyl group). The experimental work is quite thorough with appropriate control experiments. This work is ready for publication and will be of great interest to the readers of Antibiotics, but there are two points of interest that might add to the analysis:

Given that tobramycin has a long clinical history, has increased skin-darkening been noted as a side-effect?  Is there any reason to suspect that tobramycin is directly binding P38 and/or an upstream kinase? If not, given the specificity vs. the structurally-related kanamycins, is there any data on mammalian cellular targets of this class that might tease out the potential molecular target?

Reviewer 2 Report

See the attachment.

Round 2

Reviewer 2 Report

I find the answers and improvements satisfying enough.